# Transgenerational effects of alcohol on behavioral sensitivity to alcohol in *Caenorhabditis elegans*

Dawn M. Guzman[1], Keerthana Chakka[1], Ted Shi[1], Alyssa Marron[1], Ansley E. Fiorito[1], Nima S. Rahman[1], Stephanie Ro[1], Dylan G. Sucich[2], Jonathan T. Pierce[1]*

1 Department of Neuroscience, Waggoner Center for Alcohol and Addiction Research, Center for Learning and Memory, University of Texas at Austin, Austin, Texas, United States of America, 2 Department of Biology, Johns Hopkins University, Baltimore, Maryland, United States of America

* jonps@utexas.edu

**Data Availability Statement:** All raw data are included in a supplementary Excel file.

**Funding:** (DG) NIH/NIAAA Alcohol Training Grant T32 AA007471; (DG) Bruce Jones Fellowship;

## Abstract

Alcohol abuse and dependence have a substantial heritable component. Although the genome has been considered the sole vehicle of heritable phenotypes, recent studies suggest that drug or alcohol exposure may induce alterations in gene expression that are transmitted across generations. Still, the transgenerational impact of alcohol use (and abuse) remains largely unexplored in part because multigenerational studies using rodent models present challenges for time, sample size, and genetic heterogeneity. Here, we took advantage of the extremely short generation time, large broods, and clonal form of reproduction of the nematode *Caenorhabditis elegans*. We developed a model of pre-fertilization parental alcohol exposure to test alterations in behavioral responses to acute alcohol treatment (referred to in short as intoxication) in subsequent F1, F2 and F3 generations. We found that chronic and intermittent alcohol-treatment paradigms resulted in opposite changes to intoxication sensitivity of F3 progeny that were only apparent when controlling for yoked trials. Chronic alcohol-treatment paradigm in the parental generation resulted in alcohol-naïve F3 progeny displaying moderate resistance to intoxication. Intermittent treatment resulted in alcohol-naïve F3 progeny displaying moderate hypersensitivity to intoxication. Further study of these phenomena using this new *C. elegans* model may yield mechanistic insights into how transgenerational effects may occur in other animals.

## Introduction

Alcoholism is a prevalent and often devastating disease, characterized by a psychological and physical dependence on alcohol consumption. Investigations into familial patterns of alcohol dependence, as well as human twin studies and population genetic studies confirm the considerable heritability of the development of alcohol use disorder (AUD) [1–4]. However, genetic variation alone accounts for only a fraction of the heritability of this condition [5–7]. In addition to genetic information, epigenetic modifications that regulate gene expression may also be transmitted to offspring. Epigenetic modifications produce relatively stable changes in gene

Catalyst Grant from the College of Natural Sciences, UT Austin (JP). The funders had no role in study design, data collection and analysis, decision to publish, or preparation of the manuscript.

**Competing interests:** The authors have declared that no competing interests exist.

expression mediated by molecular marks without causing changes to the genetic code itself. Several types of environmental stimuli, such as stress, diet, and toxins appear to be capable of inducing epigenetic modifications, such as chromatin remodeling, DNA methylation, and small RNA activity [8–11]. In addition to altering the epigenetic landscape of the affected individual, there is increasing evidence that environmental influences can produce heritable epigenetic effects [12].

Rigorous investigation into the phenomenon of transgenerational epigenetic inheritance (TEI) has begun to take off over the last decade (e.g.[8, 10, 13–22]). It has been understood for some time now that epigenetic modifications can be brought on by environmental exposures and that the epigenetic architecture of a cell can be transmitted to its daughter cells (as in mitotic differentiated cells) [14, 23]. While it is clear that these epigenetic phenomena occur in somatic cells, the extent of epigenetic plasticity and inheritance in germ cells is still debated [15, 24]. In order for epigenetic information to be inherited by progeny, it must be transmitted through the germ cells. However, in the germ line and early embryos of many animals, including humans, these cells undergo extensive reprogramming, through which DNA is demethylated and chromatin marks are largely, if not completely removed [14, 24]. Similarly, the nematode *Caenorhabditis elegans* undergoes a large reprogramming event during early embryogenesis, during which certain H3 histone marks (H3K4me2, H3K8 and H3K18 acetylation) are dramatically reduced in germline precursor cells, while others remain stable or increase (H3K36me, H3K27me3) [24]. Therefore, any parental epigenetic "memory" that is to be passed on would have to be resistant to this reprogramming somehow, either through incomplete erasure or with the help of some maintenance mechanism. In spite of this apparent obstacle, evidence for transgenerational influence of a wide variety of environmental perturbations continue to grow steadily [24–29]. While some studies have yielded promising findings, mechanisms for stable transgenerational transmission are still a mystery.

Among the relatively few studies of TEI on alcohol, most have used rodent models to focus on F1 or F2 generations that are exposed directly to alcohol [30]. When a pregnant female rodent is exposed to alcohol, her fetus (F1) and its developing germ-line (F2) will be directly be exposed to alcohol and therefore phenotypes observed in F1 and F2 off-spring represent intergenerational effects. By contrast, the F3 generation is the first generation that is not directly exposed to alcohol, and thus represents true transgenerational epigenetic inheritance following in utero drug exposure. Nizhnikov et al., (2016) found that ethanol (EtOH) exposure of the F0 dam while pregnant with the F1 offspring increased EtOH intake by about 40% across F1, F2, and F3 generations [31]. More direct measures of acute behavioral responses to ethanol suggested TEI resistance to depressing effects of 3.5 g/kg injected EtOH. Loss of righting reflex duration was shorter and blood EtOH concentration upon awakening was higher. However, these differences from water treated control were found for F1 and F2, but not the F3 generation. A follow up study exploring parental TEI found that sires experiencing 8-day EtOH binge consumption produced progeny that displayed increased EtOH consumption in the F1 but not F2 generation relative to control [32]. These studies are important to understand if and how abuse of EtOH during fertilization and pregnancy lead to TEI, but complementary studies are needed to explore how EtOH consumption *prior* to fertilization may elicit TEI that continues to the F3 generation.

In our study, to test the transgenerational effects of ethanol (EtOH) exposure on behavioral sensitivity to EtOH, we turned to *C. elegans*. This genetically tractable nematode possesses several unique qualities that are advantageous for transgenerational studies [33]. Perhaps the most useful aspect of using *C. elegans* for any multigenerational study is its very short generation times. The duration from the parental (F0) generation laying of F1 eggs to adulthood of the F3 generation is only 12–15 days [34]. By contrast, using a mouse or rat model, the time

from fertilization of the F1 generation to sexual maturity of F3 puberty is approximately 7.5 months [35]. In addition to carrying out this multigenerational procedure in a fraction of time, each hermaphroditic worm self-crosses to produce ~200–300 offspring in just three days. This allows us to quickly propagate hundreds to thousands of individuals each generation, which is essential for collecting sufficient samples for both molecular analysis and behavioral testing. Because *C. elegans* can self-fertilize as hermaphrodites, no mating is required and working with these clonal populations allows us to confidently attribute phenotypic changes to epigenetic mechanisms. Finally, advantageous for EtOH studies, *C. elegans* displays measurable, dose-dependent behavioral responses to EtOH including reduced locomotion and postural changes [36, 37].

By exploiting these valuable qualities, we used *C. elegans* to address the question of how ethanol exposure in the parental generation affects ethanol sensitivity in subsequent generations. Knowing that AUD risk is heritable and that some of the genes known to contribute to that risk involve alcohol sensitivity and metabolism, we hypothesized that ancestral chronic treatment with EtOH would reduce EtOH-sensitivity in later generations. In other words, the EtOH line would show resistance to acute intoxication compared to the untreated-control line. The ability to pass on EtOH resistance to offspring in this circumstance may benefit *C. elegans* which feeds on bacteria in rotting vegetation that can include EtOH [38–40]. Worms resistant to intoxication may better feed on a patch of EtOH-laced food to outcompete EtOH-sensitive competitors over a few days to weeks [41]. While theoretically beneficial in the wild for worms, hypothetical transgenerational resistance may alter AUD risk in humans. Reduced acute sensitivity to alcohol in individuals was found to increase risk of developing AUD compared to those who are naively sensitive to intoxication [42, 43].

Using the measure of locomotion as a proxy for alcohol sensitivity, we assessed acute intoxication responses in the ethanol-naïve F1-F3 progeny of chronic EtOH-treated animals as well as untreated controls. We followed this up by repeating the study using an intermittent EtOH treatment paradigm. For both treatment paradigms, we found that F3 progeny in the EtOH-line displayed moderate but significant difference in EtOH sensitivity from the control line–resistance to intoxication after chronic treatment and hypersensitivity to intoxication after intermittent treatment.

## Materials and methods

### Animals

*C. elegans* was maintained as previously described [33]. Briefly, worms were raised at 20˚C on standard plates which are 6-cm-diameter Petri dishes filled with 12 mL of nematode growth media (NGM) agar seeded with 0.5 mL OP50-strain bacteria. The wild-type strain used in this study was lab N2 (Bristol, *Caenorhabditis* Genetics Center). Behavioral assays were conducted between February 23, 2016 and May 30, 2019 in our lab.

### Ethanol treatment paradigms

**Ethanol treatment.**   Age-matched F0 (parental generation) worms were harvested during late larval-stage 4 (L4) distinguished by morphology of the developing vulva (Fig 1). Worms were divided into a mock-treatment control group and an ethanol-treatment group. Progeny derived from these groups will henceforth be referred to as the control-line and ethanol-line, respectively. Treatment was modified from methods previously described [44]. Briefly, treatment plates (for both EtOH treatment and control-mock treatment) were prepared by dehydrating NGM agar plates seeded with 0.5 mL OP50 bacteria (see above) at 37˚ C with lids removed for 3.5 hours to better absorb liquid. To minimize variance in agar volume, we only

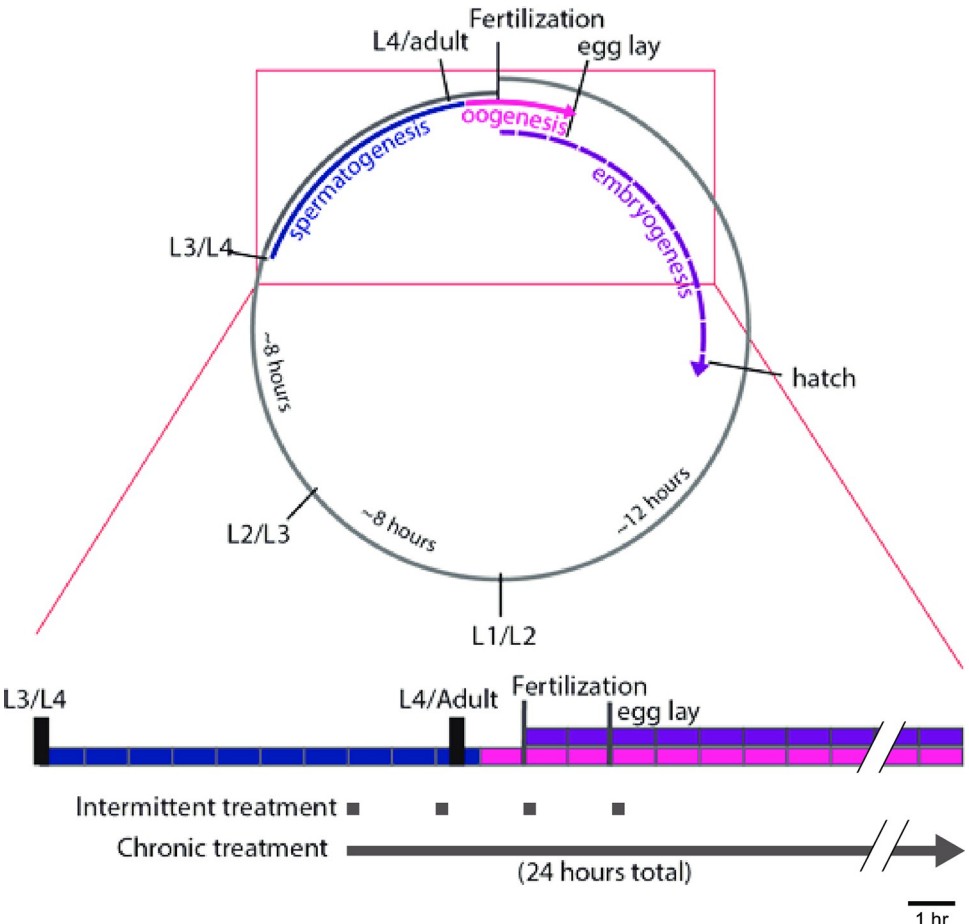

**Fig 1.** *C. elegans* **life cycle and timing of EtOH treatment.** Life cycle of an individual hermaphroditic XX female worm runs along the gray broken circle (top) from fertilization to adulthood. Black dividing lines indicate developmental milestones and molts between the four larval stages. Inset (bottom) shows linear timeline (divided into 1-hour blocks) of development starting at L4 larval stage. Spermatogenesis (blue) occurs throughout the L4 larval stage and concludes just after the L4/adult molt. Oogenesis (pink) begins in early adulthood and continues throughout adulthood. Embryogenesis of self-cross progeny (purple) begins shortly after oogenesis. In both the chronic and intermittent EtOH treatment paradigms, EtOH was introduced at the late-L4 larval stage. Worms were exposed to EtOH either intermittently or continuously for 8- or 24-hours, respectively.

used plates that after dehydration weighed 18000 ± 500 mg. The EtOH-treatment plate was then prepared by pipetting 280 μL of pure, 200 proof, Ethyl alcohol (Sigma Aldrich) beneath the agar to achieve a final EtOH concentration of 400 mM. Mock-treatment plates were prepared by similarly adding 280 μL of distilled water. EtOH-line and control-line worms were separately transferred with a platinum pick on EtOH-treatment and control plates, respectively, and sealed with Parafilm.

**Chronic ethanol treatment.** Worms were kept on their respective 400 mM EtOH- and mock-treatment plates for 24 hours that were seeded with 0.5 mL OP50 bacteria and weighed 18000 ± 500 mg. This concentration and duration of EtOH-treatment was previously shown to produce an internal EtOH concentration of ~40–60 mM, a level relevant to human consumption [44]. Both EtOH and control plates were pre-seeded with OP50 bacteria as food. Following treatment or mock treatment, the worms were then transferred to EtOH-free, OP50 seeded recovery plates, where they remained for another 24 hours before harvesting progeny.

**Intermittent ethanol treatment.** Age-matched F0 worms were harvested during late L4 stage and randomly assigned to the control-line or ethanol-line. Worms remained on the OP50 seeded EtOH treatment plates for 15 minutes, which is sufficient time period to cause intoxication as determined by locomotion and posture behavioral endpoints [36]. Next, worms were transferred to seeded recovery plates, where they remained for another 1 hour and 45 minutes. This duration is more than sufficient time for EtOH to reach undetectable levels in *C. elegans* [36, 44–46]. This treatment process was repeated three more times, for a total of four 15-minute treatments with four recovery periods following each treatment over a span of 8 hours. After the final treatment, worms were moved to a final EtOH-free seeded recovery plate for 1 hour and 45 minutes and were subsequently moved to a new seeded plate from which F1 eggs would be harvested.

**Recovery.** The recovery period durations above were chosen to allow sufficient time to achieve two goals. First, we wanted ensure that EtOH had been expelled and/or metabolized by the worm to undetectable levels, and previous studies showed that this occurs within 1-hour removal [44, 45]. Second, we wanted to wait to harvest progeny that were not directly exposed to EtOH in utero; the final 24-hour recovery period allowed F0 worms to lay fertilized eggs that may have been directly exposed to EtOH.

## Transgenerational experimental design

After the recovery window in each treatment paradigm, the worms were transferred again to new plates for 2–10 hour timed egg laying to produce the age-matched F1 generation. Early F1 adults were allowed to lay eggs to produce the F2 generation before being collected for behavioral testing. This process was repeated using F2 early adults to produce the F3 generation. The F3 generation was raised to adulthood for behavioral testing. The experimenter was blind to control or EtOH line status during set up, and blind to control or EtOH line status, baseline or EtOH treatment, and F1-F3 generation during image analysis.

## Age synchronization of animals before behavioral analysis

We performed selections to synchronize developmental age at three phases. First, each generation was allowed to lay eggs over a 2–10 hour time window. Second, we later selected midstage L4 larvae (an approximate 3 hr time window of development) using morphology of the vulva. Third, we further synchronized by selecting day 1–2 adults that harbored between 8–30 eggs. This last selection excludes adults may contain hatched larvae which would cause internal injury and prevent movement [47]. Almost all worms passed these final exclusion criteria and we did not notice an exclusion bias for any particular group.

## Measuring acute intoxication

All worms used for behavioral testing were well-fed, age-matched early (day 1–2) adults. Acute EtOH exposure reduces the speed and dampens the S-shaped posture of the worm [36]. To quantify these behavioral (intoxication) responses to control and across generations, we measured the projected area of animal bodies as viewed from above the assay plate (Fig 2A). Worms were picked from their culture plate with bacterial food to a separate unseeded plate to allow bacteria on their bodies to slough off before transferring them to the assay plate. Assay plates with any visible food were excluded. Groups of worms were confined within copper corrals on assay plates. Each group was recorded over 2-minute time windows at 1 Hz in baseline control conditions (Fig 2A1) followed by EtOH conditions (Fig 2A2). For baseline conditions without ethanol, we waited 90 to 120 seconds before recording to avoid a period when *C. elegans* suppresses spontaneous reversals after transfer to a new plate, which would increase

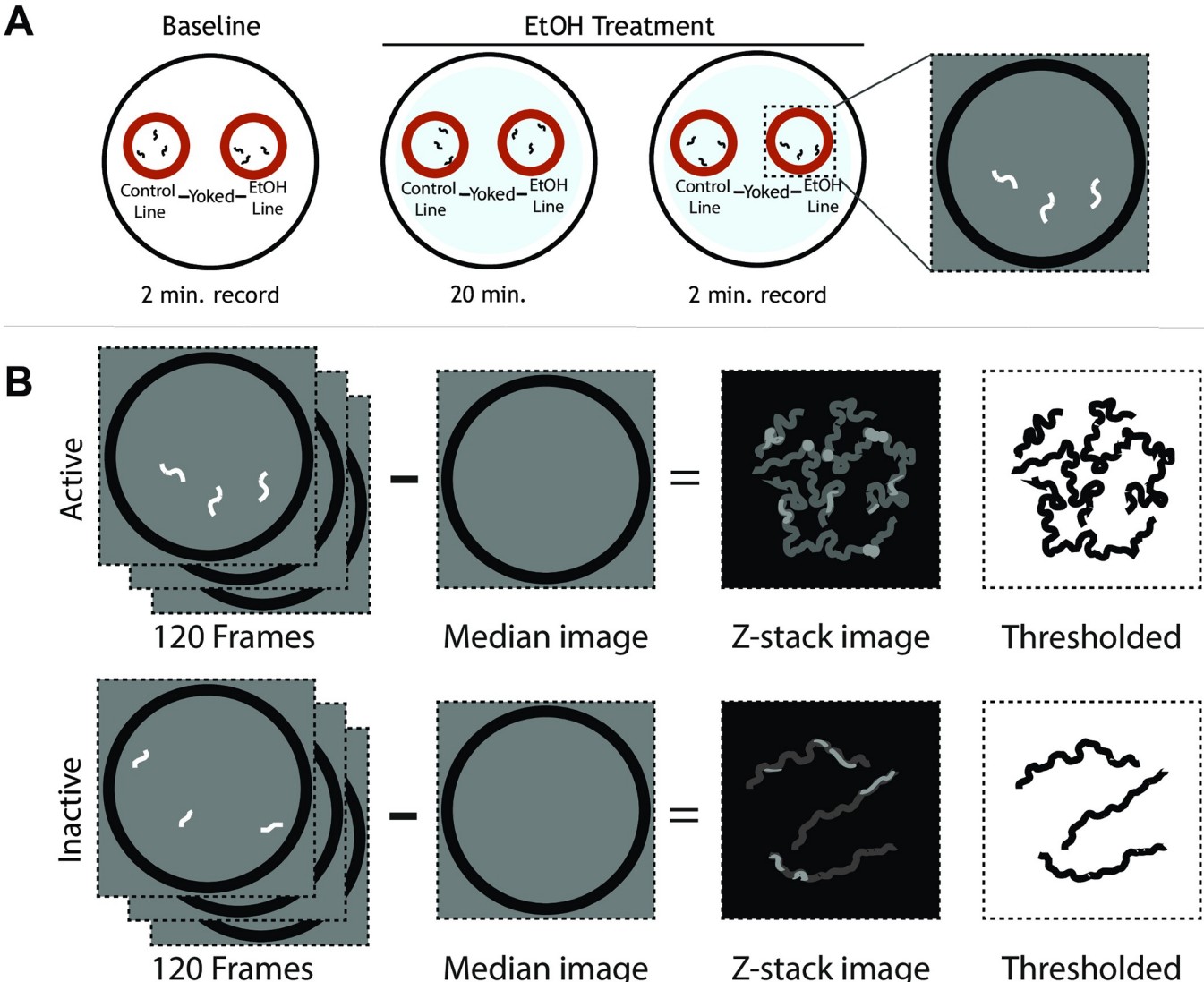

**Fig 2. Quantification of *C. elegans* ethanol sensitivity.** A. This assay measures sensitivity to acute EtOH exposure by comparing baseline and EtOH-treated locomotion. A trial consists of paired groups of 8–16 worms from control- and EtOH-line yoked on the same plate in separate copper corrals. After a 90–120 second acclimation period, the yoked groups of worms were recorded for 2 minutes at 1 Hz to measure baseline locomotion. The yoked groups were then moved to a plate containing 400-mM EtOH to incubate 20 minutes on EtOH-infused plates before being recorded again for 2 minutes. B. Image stacks of each corral were used to calculate and subtract the median background image of all 120 frames for each group to visualize the worms. Next, the stack of worm images were projected across time (Z stack) and thresholded to control for irregular lighting to display the area covered by the worms during the trial. Examples are shown for active and inactive populations of worm (above and below respectively). The extent of locomotion was quantified by calculating the percentage of the image area covered in pixels.

displacement during locomotion [48]. Worms were incubated for 20 minutes on unseeded plate infused with EtOH before recording, a duration previously demonstrated to induce intoxication [36].

In brief, Fiji software [49] was used to quantify the area worms covered while moving during the assay (Fig 2B). Untreated worms move with wide bends that generate S-shaped postures with cover a wide area, whereas intoxicated worms typically move sluggishly with narrow bends that generate flattened postures cause the worm to cover less area [36]. Area covered was calculated by generating a median background image to substract from all images in

the stack, and then projecting the maximum pixel intensity into a single image using the Z Project tool (Fig 2B). The Z-projection image was then thresholded using the IsoData auto threshold setting to account for uneven lighting for the greyscale image (Fig 2B). This final black and white image represents the area covered by worms over the entire video, in white pixels on a black background. We calculated the percentage of total area covered by worms by measuring the total number of black pixels and dividing by the total area of the field of view. This value was also divided by the number of worms that participated in the video, generating the percentage of total area covered per worm. This process was repeated for all videos.

## Statistical analyses

Data that passed Shapiro-Wilk normality test were analyzed using three-factor ANOVA, two-sided paired t-tests, or two-sided one sample t-tests and the Tukey HSD method for post hoc multiple comparisons tests with SPSS software version 28 [50].

## Results

### Ancestral chronic EtOH exposure confers modest resistance to intoxication in F3 generation

To test our hypothesis that EtOH exposure may confer transgenerational inherited naïve resistance to intoxication in *C. elegans*, we measured EtOH sensitivity in F1 through F3 generations using locomotion. We tested two lines: an EtOH line where the F0 generation was removed from food and exposed to EtOH on a plate with food, and a control line where the F0 generation was removed to a new plate with food for the same period of time. We chose a 24-hour chronic exposure paradigm with an EtOH concentration of 400 mM because it produces intoxication in the short term (20 min) and alcohol withdrawal over the long term (24 hrs) [36, 44, 45] Parental F0 worms were selected from late L4 and randomly assigned to EtOH- or control-line groups. We collected a portion of F1-F3 adult progeny from each line as described in methods and quantified their naïve sensitivity to EtOH yoked within separate corrals on the same assay plates (Fig 1A).

We evaluated EtOH sensitivity by the degree to which EtOH reduced locomotion as described previously [36, 51]. Briefly, we measured the baseline locomotion of a group of 8–16 worms (average 11.7 ± 2.74 SD) from control-line and another group from EtOH-line in adjacent yoked copper-ring corrals on foodless assay plate for a 2-minute time window. The two yoked groups of worms were next transferred to corrals on another assay plate containing EtOH for repeated measures. After a 20-minute incubation, we measured the on-EtOH treated locomotion of worms for a second 2-minute time window (Fig 1). We chose a 20-minute treatment on 400-mM EtOH because this exogenous dosage produces intoxication and results in an internal concentration of approximately 40–60 mM in *C. elegans* by previous measures [36, 44, 46]. This internal concentration is comparable to blood alcohol concentrations (BAC) observed during intoxication in humans [52]. Locomotor activity was quantified by using image analysis to determine the percent area of the corral covered per worm as they moved. In total, 194 yoked baseline and on-EtOH trials were run.

From scatter and box plots of locomotion data, we noted that intoxication was apparent in all cases because the distributions of locomotion values of EtOH-treated worms were decreased compared to corresponding baseline values regardless of generation or lineage (Fig 3A). The overall locomotion values for worms in the F3 generation were unexpectedly significantly lower than those for F1 and F2 generations. However, the lower locomotion values in

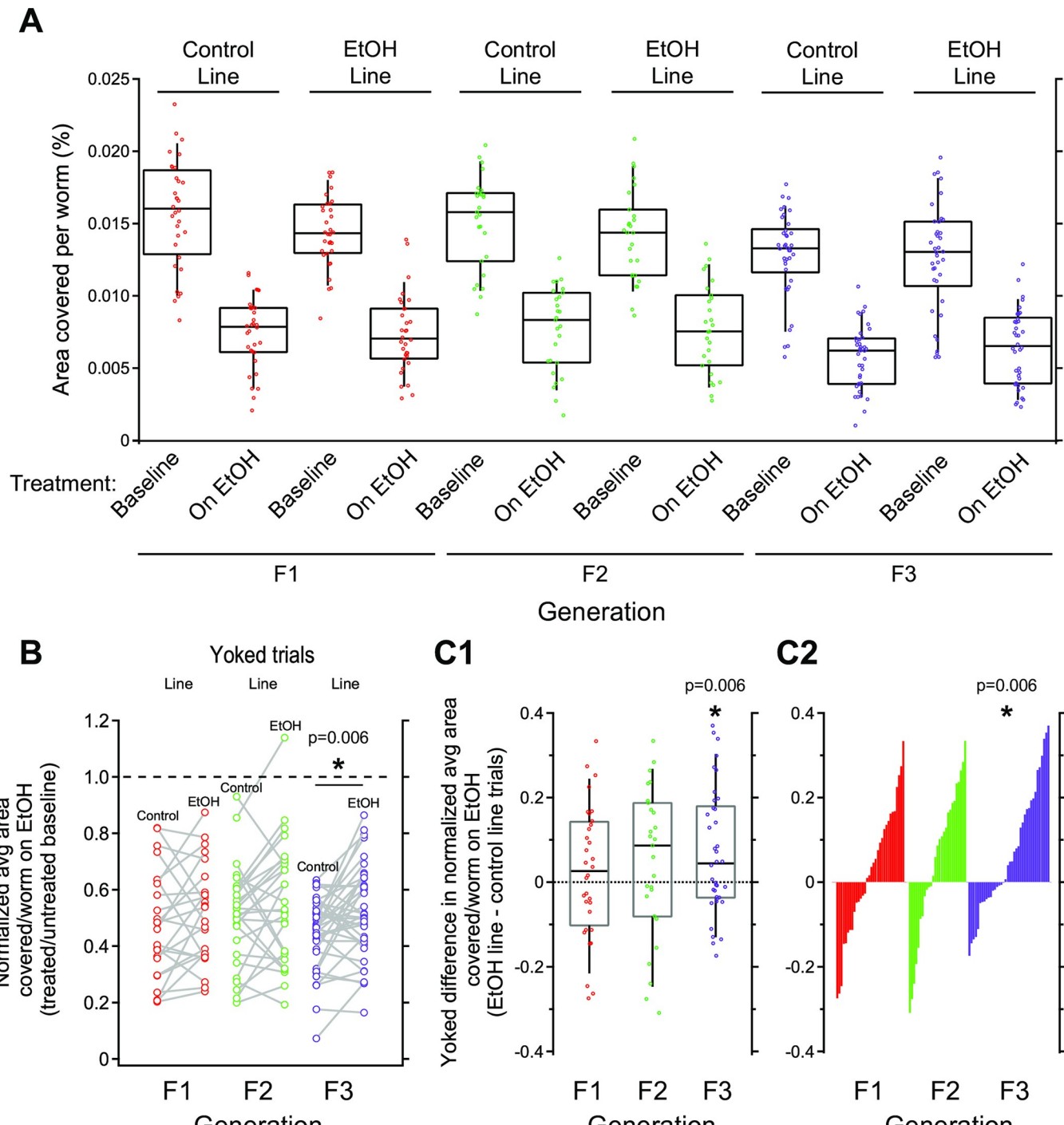

**Fig 3. Chronic EtOH-line worms show resistance to intoxication in F3 generation when yoked-lineage groups are compared.** A. Sensitivity to acute intoxication as measured by percent area covered per worm. EtOH-line animals were derived from a F0 generation that underwent chronic ethanol treatment. Each dot represents a group of 8–16 worms that was assayed to measure locomotion in untreated baseline followed by EtOH-treated conditions. Worms covered less area when treated with EtOH relative to baseline. No significant differences were observed for main effects of lineage or generation, nor after posthoc analysis of interactions between lineage and generation ([Table 1]). B. To focus on how EtOH treatment changed locomotion from baseline, we plotted locomotion values of EtOH-treated worms normalized to baseline. To show which control- and EtOH-line trials were conducted together, yoked values are linked with a gray line. A two-sided, paired t-test revealed a significant difference between normalized locomotion for control- and EtOH-line worms in F3 generation, but not F1 or F2 generations ($p = 0.006$). C. To visualize how intoxication sensitivity may differ across yoked trials, we plotted yoked difference values, equivalent to the slopes of linked gray lines in panel B. C1. Distributions of yoked difference values were not different than zero for F1 and F2, but significantly higher than zero for F3 animals (One sample t-test, $p = 0.006$). C2. Distributions from panel C1 replotted in ranked order to highlight shift towards positive values for F3 in comparison to F1 and F2 generations. A-C. Box plots represent top and bottom 25–75% and median.

**Table 1. Summary of statistical analyses for chronic EtOH treatment paradigm.**

| Analysis method | Statistical test | | *F* value | *p* value | Significant? |
|---|---|---|---|---|---|
| Comparison of raw locomotion values (Area covered per worm) Fig 3A | Three-factor ANOVA | Main effects: | | | |
| | | Generation (F1, F2, or F3) | **17.31** | *<**0.001** | **Yes** |
| | | Lineage (Control or EtOH) | 1.188 | 0.276 | No |
| | | Condition (Baseline or EtOH) | **553.5** | *<**0.001** | **Yes** |
| | | Interaction effects: | | | |
| | | Generation * Lineage | 0.789 | 0.4550 | No |
| | | Generation * Condition | 1.019 | 0.3617 | No |
| | | Lineage * Condition | 2.779 | 0.0963 | No |
| | | Generation * Lineage * Condition | 0.051 | 0.9499 | No |
| **Analysis method** | **Statistical test** | | *t* value | *p* value | Significant? |
| Comparison of normalized locomotion values (Area covered per worm on EtOH/baseline) Fig 3B | Two-sided, paired t-tests | Generation: | | | |
| | | F1 | 0.845 | 0.405 | No |
| | | F2 | 1.387 | 0.177 | No |
| | | **F3** | **2.915** | ***0.006** | **Yes** |
| **Analysis method** | **Statistical test** | | *t* value | *p* value | Significant? |
| Comparison of yoked trials (Yoked difference of normalized area covered per worm) Fig 3C1 & 3C2 | One-sample, two-sided t-test against hypothesized mean, µ = 0 | Generation: | | | |
| | | F1 | 0.845 | 0.405 | No |
| | | F2 | 1.387 | 0.177 | No |
| | | F3 | **2.915** | ***0.006** | **Yes** |

F3 were independent of condition and lineage, suggesting that this effect was unrelated to potentially heritable or acute effects of EtOH treatment for the F3 generation.

These observations were backed up by statistical comparisons using a three-factor ANOVA (Table 1). We found a highly significant effect for condition (baseline vs. EtOH) and generation (F1, F2 or F3), but not lineage. Using posthoc pairwise comparisons, we found that locomotion data were significantly lower for F3 generation than F2 and F3 (Tukey's HSD test F1 vs.F3, *p* = 2.30 E-06 and F2 vs.F3, *p* = 3.46 E-07 respectively). Counter to our hypothesis, EtOH sensitivity was not obviously different for any particular generation, lineage, or their combination. Consistent with this observation, no significant interaction was found between generation, lineage and condition (Table 1). This initial analysis suggested that if any transgenerational effects of EtOH occurred, it may be modest and obscured by variability in our raw behavior data.

Behavioral variation in EtOH studies is typically controlled for by normalizing EtOH response to baseline values (e.g, [36, 51]). Each group of worms was assayed in baseline followed by on-EtOH conditions. To normalize data across these repeated measures, we divided the EtOH trial measure by the corresponding baseline trial measure to generate a new EtOH sensitivity value that represents a fraction of baseline for each group. These normalized locomotion values are plotted in Fig 3B with yoked control- and EtOH-line trials linked by gray lines. A value of 1 represent no change in on-EtOH locomotion from baseline, whereas values less than 1 represent depressing effects of intoxication. Analysis of normalized values for control and EtOH lines showed that locomotion values were significantly higher for EtOH-line in

F3 generation, but not for F1 and F2 generations (two-sided paired t-tests; F1: *t* = 0.845, *df* = 31, *p* = 0.405; F2: *t* = 1.387, *df* = 26, *p* = 0.177; F3: *t* = 2.915, *df* = 37, *p* = 0.006). The size of significant difference between control and EtOH line worms the F3 generation was medium as quantified by Cohen's d value of 0.473. We conclude that worms from the EtOH-line displayed significant relative resistance to intoxication when compared to yoked control-line trial counterparts (Fig 3B and Table 1).

To better visualize the effect, we plotted difference values of normalized locomotion between EtOH-line and yoked control-line trials (Fig 3C). This value represents how much the normalized locomotion of a group of worms in the EtOH-line differs from a yoked group in the control-line. The value also is equivalent to the slope of the line connecting yoked values in Fig 3B. These values are depicted in scatter and box plots in Fig 3C1 and replotted in a ranked order chart in Fig 3C2 to better observe deviations from zero. Although we already tested for significant differences between control- and EtOH-line normalized values using paired t-tests for data in Fig 3B, for demonstration purposes we also performed a different but equivalent two-sided one-sample t-test on data in Fig 3C. The distribution of values for the distribution mean was significantly higher than zero for the F3 generation (mean = 0.0698, N = 37, SD = 0.14750, $t_{(1,37)}$ = 2.915, p = 0.006), but not F1 and F2 generations in Fig 3C1 (Table 1). This is more apparent as a noticeable asymmetric right shift from zero for F3 data in the ranked order chart in Fig 3C2.

## Ancestral intermittent EtOH exposure confers hypersensitivity to intoxication in F3 generation

Finding moderate but significant transgenerational naïve resistance in F3 generation after chronic EtOH treatment made us wonder if an alternate treatment paradigm may produce a stronger or alternate effect. There are several variables in treatment schedule that we suspected may contribute to transgenerational effects of EtOH, including developmental timing of treatment (e.g. adolescent vs. adult), duration of treatment (acute vs. chronic) and frequency of treatment (single continuous treatment vs. intermittent treatment). Therefore, for our second approach we chose to investigate the transgenerational effects of intermittent EtOH-treatment on EtOH sensitivity. The treatment schedule differed from the chronic treatment in that, instead of a continuous 24-hour long exposure, worms were put on 400-mM EtOH treatment plates for 15 minutes every two hours for a total of 4 treatments (Fig 1). The 15 minute EtOH treatment is sufficient to induce intoxication in naïve worms (Davies et al., 2003), and the 1 hour 45 minutes is long enough for worms to recover from intoxication and have internal EtOH reach undetectable levels [44, 46].

Another important aspect of this intermittent paradigm that differs from the chronic paradigm is the effect on worm development during treatment. Chronic EtOH treatment, as we carried it out, starts at the late L4 larval stage (several hours before adult stage) and continues for 24 hours. Because EtOH decreases pharyngeal pumping rate, feeding is reduced for the EtOH-line worms during this 24-hour period. Growth of F0 parental worms was also noticeably stunted as a result (although locomotion measures of F1 were remarkably similar for control- and EtOH-line F1 progeny (Fig 3A)). This introduces two additional disparities between the EtOH- and control-line. By limiting the amount of continuous time spent on EtOH, an intermittent treatment paradigm may reduce the hypothetical impact of EtOH on feeding and growth.

Analysis of worms in this intermittent paradigm using raw values for locomotion represented by area covered per worm mirrored our analysis for chronic paradigm above as determined by three-factor ANOVA. We found that in all cases, worms displayed intoxication

because raw values for locomotion were significantly lower for EtOH versus baseline (Fig 4A, Table 2). Likewise, as with the chronic EtOH paradigm above, there were no significant main effects of generation or lineage, nor significant interaction effects (Table 2). Posthoc, Tukey's HSD pairwise comparison revealed a weak significant difference between locomotion values between F2 and F3 generation that were unrelated to EtOH treatment and lineage. This suggested again that if a transgenerational effects of intermittent EtOH lineage occurred, it may be weak and masked by variability in raw locomotion values.

To control for this variability, we normalized on-EtOH locomotion values to baseline values (Fig 4B). As for data from our chronic EtOH paradigm test, we found a significant difference between control and EtOH-line for the F3 generation, but not the F1 and F2 generations (two-sided paired t-tests; F1: $t = 1.390$, $df = 22$, $p = 0.178$; F2: $t = 0.736$, $df = 5$, $p = 0.495$; F3: $t = 2.789$, $df = 6$, $p = 0.032$). In contrast to our chronic EtOH treatment paradigm above, however, rather than resistance the EtOH-line worms displayed relative hypersensitivity to EtOH because they had lower values of normalized locomotion for each yoked pair (Fig 4C). Although our sample size of F3 yoked trials was small (n = 7 of two assays each), the size of effect was large as quantified by Cohen's d value of 1.05.

To better visualize the difference in normalized values, we again plotted yoked trial differences in normalized EtOH sensitivity for each generation (Fig 4C1 and 4C2). Again for demonstration purposes, we also performed a two-sided one-sample t-test against hypothesized mean of zero for each generation in Fig 4C even though this test is equivalent to the paired t-tests of data in Fig 4B. We found that all seven of these measures were less than zero (mean = -0.133, n = 7, SD = 0.127, $t_{(1,6)} = -2.789$, $p = 0.032$). We conclude that worms from the EtOH-line displayed significant relative hypersensitivity to intoxication when compared to yoked control-line trial counterparts (Fig 4C and Table 2).

## Discussion

Here we present the first study, to our knowledge that examines the effects of parental pre-fertilization ethanol exposure in the subsequent F1-F3 generations in any animal. Although similar studies were carried out in rats (testing F1-F3, also), the parental exposure was limited to only a brief prenatal (post fertilization) treatment in mothers [31] and for 8 days of binge-like consumption in fathers [32]. In our approach, the parental chronic EtOH treatment, as well as recovery from EtOH, takes place prior to fertilization of the embryos that give rise to the F1-F3 individuals later tested. Thus, the findings from our treatment paradigm are unique in that they are the result of parental somatic cell and germline cell exposure only. This represents a valuable approach because any epigenetic alterations transferred to the progeny would have resulted from some mechanism that persisted from the time of EtOH treatment to post-fertilization.

We first tested our hypothesis that chronic treatment with EtOH in the parental (F0) generation would lead to a resistance to acute EtOH intoxication in subsequent generations. Our results revealed a moderately significant resistance to intoxication between the EtOH and control lines in the F3 generation, but not F1 and F2 generations. When we repeated this transgenerational study using an intermittent EtOH-treatment schedule in the F0 generation we saw a moderately significant hypersensitivity to intoxication between the EtOH and control lines in the F3 generation, but not F1 and F2 generation. We can only speculate reasons underlying the opposite results. TEI resistance to intoxication after chronic EtOH exposure may reflect how other organisms temporarily develop resistance to persistent environmental stress or toxins [53]. TEI hypersensitivity to intoxication after intermittent EtOH exposure might be harder to understand. It may correlate with heightened sensitivity to EtOH that may

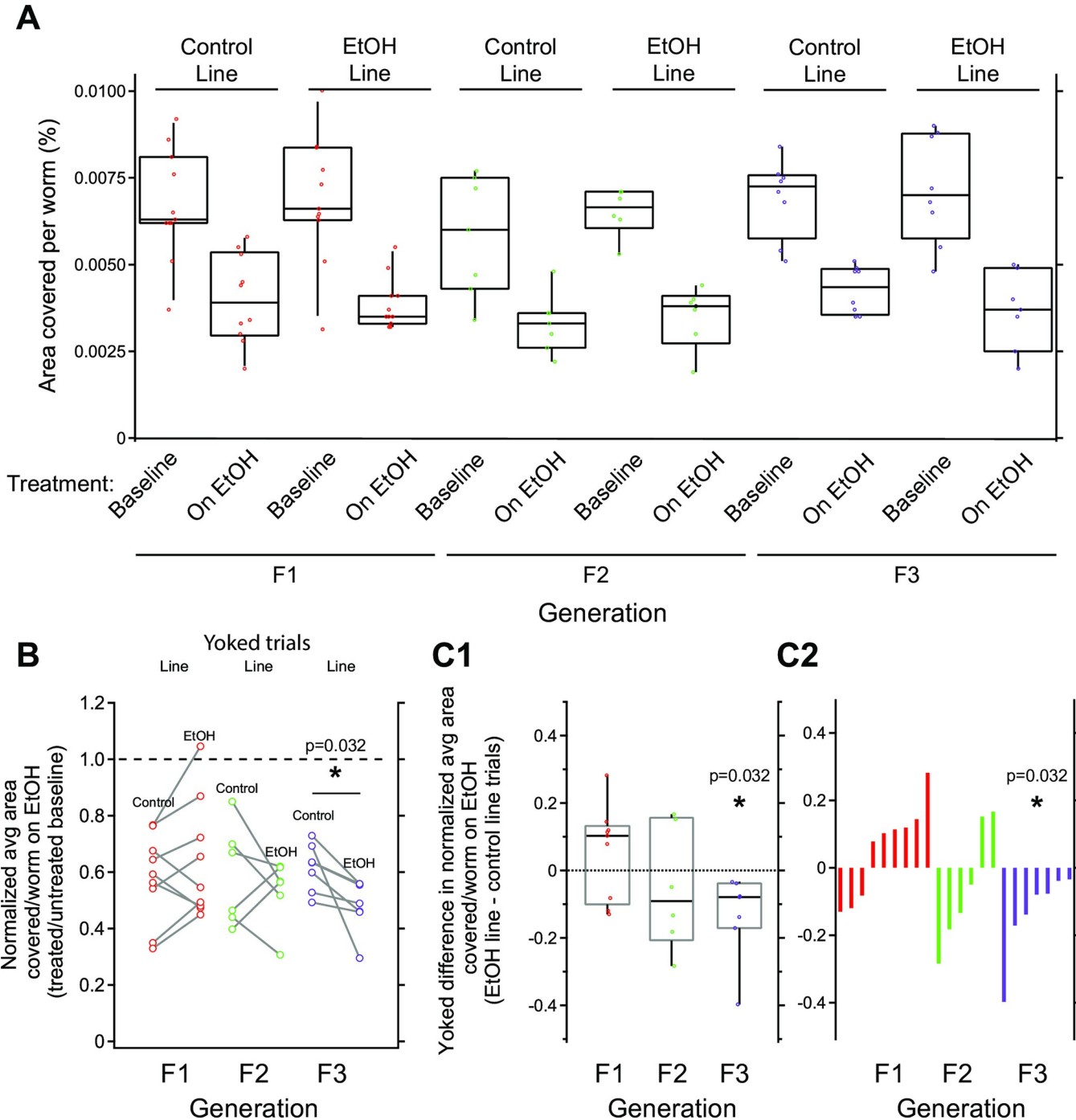

**Fig 4. Chronic EtOH-line worms show hypersensitivity to intoxication in F3 generation when yoked-lineage groups are compared.** Data are organized the same as in Fig 3. A. Sensitivity to acute intoxication as measured by percent area covered per worm. Worms generally covered less area when treated with EtOH. No significant differences were observed for main effects of lineage or generation, nor after posthoc analysis of interactions between lineage and generation (Table 1). B. Locomotion values of EtOH-treated worms normalized to baseline. Yoked values are linked with a gray line. A two-sided, paired t-test revealed a significant difference between normalized locomotion for control- and EtOH-line worms in F3 generation, but not F1 or F2 generations (*p* = 0.032). C1. To visualize differences between yoked control- and EtOH-line trials, we plotted yoked difference values. These distributions were not different than zero for F1 and F2, but significantly lower than zero for F3 animals (One sample t-test, p = 0.032). C2. Distributions from panel C1 replotted in ranked order to highlight shift towards n values for F3 in comparison to F1 and F2 generations. A-C. Box plots represent top and bottom 25–75% and median.

**Table 2. Summary of statistical analyses for intermittent EtOH treatment paradigm.**

| Analysis method | Statistical test | | F value | p value | Significant? |
|---|---|---|---|---|---|
| Comparison of raw locomotion Values (Area covered per worm) Fig 4A | Three-factor ANOVA | Main effects: | | | |
| | | Generation (F1, F2, or F3) | 2.572 | 0.082 | No |
| | | Lineage (Control or EtOH) | 0.101 | 0.752 | No |
| | | Condition (Baseline or EtOH) | **122.6** | **<0.001** | **Yes** |
| | | Interaction effects: | | | |
| | | Generation * Lineage | 0.373 | 0.690 | No |
| | | Generation * Condition | 0.106 | 0.900 | No |
| | | Lineage * Condition | 1.213 | 0.274 | No |
| | | Generation * Lineage * Condition | 0.108 | 0.898 | No |
| **Analysis method** | **Statistical test** | | **t value** | **p value** | **Significant?** |
| Comparison of normalized locomotion values (Area covered per worm on EtOH/baseline) Fig 4B | Two-sided, paired t-tests | Generation: | | | |
| | | F1 | 0.631 | 0.542 | No |
| | | F2 | 0.736 | 0.495 | No |
| | | **F3** | **2.789** | ***0.032** | **Yes** |
| **Analysis method** | **Statistical test** | | **t value** | **p value** | **Significant?** |
| Comparison of yoked trials (Yoked difference of normalized area covered per worm) Fig 4C1 & 4C2 | One-sample, two-sided t-test against hypothesized mean, μ = 0 | Generation: | | | |
| | | F1 | 0.631 | 0.542 | No |
| | | F2 | 0.736 | 0.495 | No |
| | | F3 | **2.789** | ***0.032** | **Yes** |

encourage animals to escape toxic patches EtOH in the environment. If this adaptive behavior was enhanced via TEI, we could not observe it in our experiments that used a uniform concentration of EtOH spread throughout the assay plate.

The delayed and opposite TEI effects we found may have been unexpected, but they are not unheard of. Other TEI studies have found delayed or distinct effects in each generation. One study on the transgenerational effects of endocrine-disrupting chemicals in F1 and F3 generations and beyond found several examples of behaviors affected rats in the F3 but not the F1 generation (see Figs 4 and 6 of Gillette et al., 2022) [54]. Two additional TEI studies on endocrine-disrupting chemicals in rats found that each generation displayed a different phenotype in careful head-to-head-to-head comparisons of F1, F2, and F3 [55, 56]. Another example in fish compared F0, F1, and F2 (equivalent to F1, F2 and F3 in rodents) found "emergent" effects of exposure in the F2 generation [57]. Such delayed and different effects in TEI studies are becoming more commonly reported and likely explained by differences in timing and amount of exposure (adult vs. germline vs. unexposed epigenetic inheritance. Different treatment paradigms, such as chronic versus intermittent in our study, raise the potential for triggering unique combinations of epigenetic mechanisms that may explain these different or opposite TEI effects.

The differential sensitivity to intoxication in the F3 generation between EtOH and control lines was not apparent in both treatment paradigms unless we controlled for inter-experiment variation. We found that we needed to normalized EtOH-treated locomotion measures to baseline, and also compare yoked EtOH and control line measures. We make three conclusions from these findings. First, the overall transgenerational effect for F3 generations here are

small compared to the level of variability in behavior, even baseline levels. Considering the lack of significant effects in F1 and F2 generations, we speculate the possibility that small yet significant effects in F1 and F2 generation, similar to those observed in the F3 generation, might be masked by this high behavioral variability. Second, detecting potential subtle transgenerational effects may similarly benefit from a) experimental designs that simultaneously assay control- and experimental treatment-lines on the same assay plates in parallel and also b) normalization of experimental measures to baseline to control for variability. Supporting this idea, a similar experimental design using yoked groups of worms for comparison using a higher order behavioral index revealed a transgenerational effect on memory in *C. elegans* [58, 59]. Although these results modestly support our hypothesis of resistance to acute intoxication in the EtOH-line for chronic exposure, this is most likely not the end of the story. TEI can be influenced by any number of factors, and not necessarily our experimental treatment alone [8–11]. Even in our relatively well-controlled laboratory setting, variables beyond parental EtOH-treatment likely influenced transgenerational changes in behavior and gene expression. These include starvation, treatment schedule and seasonal conditions, which are elaborated below. Third, we speculate that the small effects in F3 (and possibly effects potentially masked in F1 and/or F2) might be enhanced if the TEI experimental design was controlled further. For instance, although most assay plates were selected to have the same volume, the inclusion criteria was set to ±2.7% of weight before adding a fixed volume of EtOH. To achieve more uniform EtOH content across plates, future studies may consider tightening this criterion as well as adjusting the volume of EtOH added to each plate based on its weight.

An unintended consequence of our chronic EtOH-treatment paradigm is starvation. The intoxicating dose of EtOH reduces the rate of feeding by directly inhibiting pumping of its pharynx, a bi-lobed pump that sucks in bacteria as food via peristalsis. In turn, this dramatically decreases food intake. We began the chronic treatment during the L4 larval stage of development. During this time period the worm normally grows by 25% in length and around 40% in volume as it enters the adult stage [60, 61]. Although we have not quantified it, growth was stunted in the EtOH-line F0 worms following chronic EtOH treatment. A recent worm study investigated the intergenerational effect of maternal dietary restriction (DR) by raising DR worms in liquid culture containing low concentrations of the E. coli food source. This dietary restriction results in smaller early brood sizes but individual progeny were larger and were later resistant to the effects of starvation [62]. Another recent worm study invested the transgenerational effects of starvation. In that study, the F0 generation was exposed to prolonged starvation (8 days) during the L1 larval stage, which leads to "larval arrest." When the F0 were allowed to recover from starvation, they took longer to grow to adulthood, had reduced adult size and reduced fecundity [63]. Progeny of these starved F0 also had reduced embryo quality, small brood size, and their progeny were smaller, in contrast with the study described above. Based on our casual observations (not quantified), our F0 EtOH-line may have had smaller brood sizes as well. This will need to be quantified in the future. We did not note a difference in size of F1 progeny, but measurements of F1 individuals could be taken from videos of acute intoxication assay. In future studies, the effects of starvation alone on F0 worms and their progeny needs to be examined by running a starved control line alongside the non-starved control line and EtOH line. A link between transgenerational inheritance of ethanol phenotypes and development phenotypes has also been observed in rats. Nizhnikov et al., (2016) found that exposing F1 offspring to EtOH in utero not only increased EtOH intake across F1, F2, and F3 generations, but also decreased weight of F2 and F3 generations [31].

In addition to our chronic EtOH-treatment study, we investigated the transgenerational effects of intermittent EtOH-treatment. As with our chronic EtOH-treatment, we expected this treatment to result in resistance to acute intoxication in subsequent generations. Also, if

starvation did interfere with the results of the chronic EtOH-treatment study, this approach should have reduced that interference. In contrast to our findings for chronic EtOH treatment, the effect showed increased sensitivity in the EtOH-line in the F3 generation. These unexpected results warrant further investigation.

Besides investigating heritable phenotypes that result from ancestral EtOH-treatment, gene expression changes also need to be surveyed. We are particularly interested in looking at changes in *slo-1* expression. Through two saturating forward genetic screens, the *slo-1* gene found to be a major target for EtOH is required for acute intoxication in *C. elegans* [36]. This gene encodes the BK channel, a large-conductance calcium- and voltage-activated potassium channel. Worms lacking this channel show very strong resistance to intoxication, show more severe EtOH withdrawal-related impairments [44], and it is a conserved target for EtOH that is also found in mouse, fly and humans [64–66]. The resistance to intoxication of worms lacking this channel make *slo-1* an appealing candidate gene to investigate in worms following ancestral EtOH treatment. A broader survey of gene expression changes following parental EtOH-treatment may inform future endeavors in investigating behavioral transgenerational effects of EtOH.

There are several recent studies using *C. elegans* that utilize environmental triggers in their approach to study TEI. As described above, starvation and dietary restriction have been used as an environmental trigger for TEI [62, 63]. Both of these studies uncovered effects of starvation or dietary restriction that involved, growth, brood size, and stress resistance phenotypes. Other environmental triggers that have been investigated in *C. elegans* are pathogenic bacteria exposure [58] and exposure to diesel particulate matter, an industrial air pollutant [67]. In the former, when worms learned to avoid pathogenic bacteria *Pseudomonas aeruginosa* (PA14) in the F0 generation, progeny that had never encountered the bacteria showed avoidance for four generations (F1-F4). Inheritance of this pathogenic avoidance phenotype could be prevented by disruption of the piRNA pathway.

The recent study that investigated the transgenerational effects of diesel particulate matter (DPM) on TEI [67]. They found that a 24-hour exposure to 1.0 µg/mL DPM in the parental (F0) generation during the L4 larval stage resulted in increased germ cell apoptosis in these animals, as well as reduced brood size. Germ cell apoptosis was assessed in the F1-F5 generations following F0 exposure. A less pronounced increase in germ cell apoptosis was seen in the F1 generation, but by the F2 generation, the number of apoptotic germ cells had recovered to the level seen in control (unexposed) animals. The increase in apoptotic germ cells in F1 did not occur when F0 was exposed to a lower concentration (0.1 µg/mL) of DPM. The researchers also carried out another transgenerational study in which worms were exposed to DPM in consecutive generations, which they referred to as "continuous" exposure. In the continuous exposure experiments, each generation, F0 and F1-F5, was exposed to DPM for 24 hours during the L4 larval stage. This resulted in a similar increase in apoptotic germ cells in the adults of each generation, with no difference between generations. The continuous exposure did, however, lead to a gradual decrease in brood size starting in the parental generation and continuing to decrease each generation through to F5. Unlike the pathogenic bacteria avoidance study, which provided strong evidence for true transgenerational epigenetic inheritance by an environmental trigger, these results more likely demonstrate a case of consecutive intergenerational effects. If DPM exposure leads to reduced brood size and increased germline apoptosis in the exposed generation and immediate progeny, it is feasible that consecutive or continuous exposures would have an additive effect, wherein the toxin leads to reproductive decline across generations.

Although the environmental triggers used to induce TEI differ between these studies and also differ from our own, there are some commonalities to the approaches used. For example,

in several of the studies, exposure to the trigger took place during a developmental window in the parental generation that may be vulnerable to epigenetic perturbation. Pathogenic bacteria avoidance training was carried out during the L4 larval stage, as well as DPM exposure and our EtOH treatment [58, 67]. The *C. elegans* germline is developing during this stage, and importantly, spermatogenesis takes place and is completed just as the worm transitions from L4 to young adult. This means that when a trigger is initiated during the L4 stage and persists for 24 hours (as was the case in all three of these studies), the entire lifetime supply of sperm cells the worm carries would have been exposed to the trigger. Besides the developmental timing (i.e. exposure during L4 stage), the duration of the trigger may be a key factor in TEI. This time window and 24-hour duration of exposure successfully induced transgenerational avoidance of pathogenic bacteria, but the researchers did not see inheritance of this behavior with a shorter (4-hour) avoidance training in the parental generation. This timing and duration of the parental trigger also led to intergenerational inheritance of reproductive impairment following DPM exposure.

More and more studies are leveraging the powerful traits of *C. elegans* to study TEI in different contexts. Although we only observed modestly significant heritable phenotype in our study using this timing and duration of alcohol treatment, future studies may find ways to enhance this phenomenon by lowering variability. *C. elegans* continues to represent a promising approach for transgenerational research.

## Supporting information

**S1 Data.**
(XLSX)

## Acknowledgments

We would like to thank Dr. Vishy Iyer for originally approaching us with the idea of a transgenerational study using *C. elegans*. We would also like to thank Dr. Andrea Gore for intellectual contributions, as well as Susan Rozmiarek and Cory Gentry for expert assistance.

## Author Contributions

**Conceptualization:** Dawn M. Guzman, Dylan G. Sucich, Jonathan T. Pierce.

**Data curation:** Dawn M. Guzman, Jonathan T. Pierce.

**Formal analysis:** Dawn M. Guzman, Jonathan T. Pierce.

**Funding acquisition:** Jonathan T. Pierce.

**Investigation:** Dawn M. Guzman, Keerthana Chakka, Ted Shi, Ansley E. Fiorito, Nima S. Rahman, Stephanie Ro.

**Methodology:** Dawn M. Guzman.

**Project administration:** Jonathan T. Pierce.

**Supervision:** Jonathan T. Pierce.

**Visualization:** Alyssa Marron, Jonathan T. Pierce.

**Writing – original draft:** Dawn M. Guzman, Jonathan T. Pierce.

**Writing – review & editing:** Alyssa Marron, Jonathan T. Pierce.

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
