## [Decision Letter · Decision Letter 0]

5 Sep 2022

PONE-D-22-19245Transgenerational effects of alcohol on intoxication sensitivity in Caenorhabditis elegansPLOS ONE

Dear Dr. Pierce,

Thank you for submitting your manuscript to PLOS ONE. After careful consideration, we feel that it has merit but does not fully meet PLOS ONE’s publication criteria as it currently stands. Therefore, we invite you to submit a revised version of the manuscript that addresses the points raised during the review process.

Please address the methodological concerns of Reviewer #1. Please also make appropriate text modifications to address the question of the generational delay in seeing an effect.

I am somewhat concerned that the results here are due to reasons other than epigenetic effects, such as, for example, some of the methodological concerns of Reviewer #1. The effects are _extremely_ subtle, they are only seen in the F3 in contrast to other TEI studies, and the two treatment paradigms yielded opposite phenotypes, when I think that they would be expected to agree. I believe that these concerns can be addressed with careful explanations in the text. 

We look forward to receiving your revised manuscript.

Kind regards,

Jill C Bettinger, Ph.D.

Academic Editor

PLOS ONE

Journal Requirements:

"We would like to thank Vishy Iyer for originally approaching us with the idea of a transgenerational study using C. elegans, and for collaborating with us on the Catalyst Grant from the College of Natural Sciences, UT Austin that provided some of the funding for this project. Additional funding was provided by the NIH/NIAAA Alcohol Training Grant T32 AA007471, the Bruce Jones Fellowship, and generous support from Tom Calhoon. We would also like to thank Susan Rozmiarek and Cory Gentry for expert assistance."

"(DG) NIH/NIAAA Alcohol Training Grant T32 AA007471; (DG) Bruce Jones Fellowship; Catalyst Grant from the College of Natural Sciences, UT Austin (JP). The funders had no role in study design, data collection and analysis, decision to publish, or preparation of the manuscript. "

Reviewers' comments:

Reviewer's Responses to Questions

**Comments to the Author**

1. Is the manuscript technically sound, and do the data support the conclusions?

Reviewer #1: Partly

Reviewer #2: Partly

2. Has the statistical analysis been performed appropriately and rigorously? 

Reviewer #1: Yes

Reviewer #2: Yes

3. Have the authors made all data underlying the findings in their manuscript fully available?

Reviewer #1: Yes

Reviewer #2: Yes

4. Is the manuscript presented in an intelligible fashion and written in standard English?

Reviewer #1: Yes

Reviewer #2: Yes

5. Review Comments to the Author

Reviewer #1: Guzman et al describe two experiments designed to examine the potential for ethanol to generate transgenerational effects on behavioral responses to the effect of ethanol on locomotion in later generations. The development of a protocol to detect true trans-generational effects of the drug would be highly significant, particularly in combination with a genetically tractable organism such as C. elegans. The authors employ two ethanol exposure paradigms, one supported by the literature for generating developmental and withdrawal-associated effects, and the other, an intermittent exposure that is used to minimize potential effects of ethanol on feeding and growth. They examine F1, F2 and F3 generations for altered responses to ethanol, and only detect changes in the F3 generation under each ethanol exposure paradigm. The F3 generation of the chronically-exposed animals have a degree of ethanol resistance relative to paired control animals, whereas the F3 generation of the intermittently-exposed P0 animals have a degree of ethanol hypersensitivity. The effects are subtle enough that statistical differences between worms with an ancestral history of ethanol exposure and those without are only detectable when variation in baseline and day to day variation in behavioral response is considered. The data from each experiment are analyzed in several ways to illustrate the proposed differences in the F3 generation animals.

One issue of significant concern is that effects are only detected in the F3 generation, rather than in the F1, F2 and F3 generations as might be expected with a putative epigenetic effect. That they are not seeing changes in the earlier generations might be considered supporting evidence for these not being epigenetic effects related to the ethanol exposures. The authors should provide reasonable speculation or evidence from the literature to explain why the observed changes in behavioral responses to ethanol are only seen in the F3, not in the F1 and F2 generations, which presumably are also bearing a similar epigenetic change if it was induced in the oocytes (and possibly sperm) of the P0.

I have several methodological queries and points requiring more detail that are very important for understanding where some of the behavioral variation may be occurring.

1. There appears to be an assumption that the volume of media in the petri plates is identical for every plate, such that 280 µl of ethanol is added to every plate to achieve a final ethanol concentration of 400 mM. If there is any variation in the volume of media, then different plates will have different ethanol concentrations. How are the authors ensuring that the volume of media is the same for each plate? Even if a very accurate method of adding media to the plates is used, media in petri plates often dehydrate over time and at different rates depending on the laboratory conditions and their method of storage. The methods need to be expanded to describe this.

2. Related to point 1, is the volume of OP50 culture added to each plate the same? If not, then different plates will have had different volumes of liquid added prior to the 37°C dehydration step.

3. The length of time used for the timed egg-laying to produce age-matched progeny should be specified. A shorter window of egg laying will lead to animals that are more closely age-matched.

4. Related to point 3, the animals used for behavioral testing are described as age-matched day 1-2 adults. There is a significant difference in size of day 1 and day 2 adult worms. Are the worms being compared specifically day 1 adults or day 2 adults or is there a mixture of ages? Importantly, size differences between animals could lead to issues with the method used to track locomotion if body size was not controlled as locomotion is being measured based on area coverage over time.

5. For the baseline locomotion measurement, the time for how long the worms have been on the plate before the movie of their movement was recorded should be specified. If the recording is immediate, is there concern that the measurement is including the reaction of the worms to just having been moved?

6. If the worms are moved from seeded plates to unseeded plates for the behavioral assays, are the worms moved from those seeded plates without transferring any bacteria? If so, how? Bacteria that are transferred with the worms will affect the locomotion pattern of worms on a plate that lacks bacteria elsewhere.

7. Why are there different numbers of 1st (n=32), 2nd (n=27), and 3rd (n=37) generation yoked trials? This is also the case for the intermittent exposure. The methods suggest that behavioral testing was performed at each generation of a lineage, so there should be an equal number of trials across the generations. Also, the text references N=38 for 3rd generation analyses but only 37 trials are listed in the supplementary data for F3 trials.

Minor issues:

line 81 “a” mystery

line 88 EtOH should be written out in full when first used

line 125 “may be benefit”, delete “be”

line 271 “trial” should be plural

line 430 perhaps add the word “it” to “we could not observe (it) in our experiments”

Throughout: “fertilization” should be used rather than “conception”

Reviewer #2: This is an interesting study describing the transgenerational impact of prior ethanol exposure on locomotion in subsequent generations. While the findings are interesting, the characterization of these findings is very limited. In particular, the metric of area covered is not detailed enough to truly represent intoxication. Other metrics could be measured such as average speed or amplitude or many other measurements of locomotory behavior. The authors often reference published work but have not done the work in this particular study to truly show that the worms are indeed being intoxicated. Even the term intoxicated is vague and should be replaced by a more apt term. Finally, Figure 2 is low quality and should be redone.

6. PLOS authors have the option to publish the peer review history of their article (what does this mean?). If published, this will include your full peer review and any attached files.

Reviewer #1: No

Reviewer #2: No

---

## [Author Response · Author response to Decision Letter 0]

13 Sep 2022

PONE-D-22-19245

Transgenerational effects of alcohol on intoxication sensitivity in Caenorhabditis elegans

PLOS ONE

Dear Dr. Pierce,

Thank you for submitting your manuscript to PLOS ONE. After careful consideration, we feel that it has merit but does not fully meet PLOS ONE’s publication criteria as it currently stands. Therefore, we invite you to submit a revised version of the manuscript that addresses the points raised during the review process.

Please address the methodological concerns of Reviewer #1. Please also make appropriate text modifications to address the question of the generational delay in seeing an effect.

I am somewhat concerned that the results here are due to reasons other than epigenetic effects, such as, for example, some of the methodological concerns of Reviewer #1. The effects are _extremely_ subtle, they are only seen in the F3 in contrast to other TEI studies, and the two treatment paradigms yielded opposite phenotypes, when I think that they would be expected to agree. I believe that these concerns can be addressed with careful explanations in the text. 

>>Thank you for seeing merit in our study. We agree that the effects we found are small, yet consistent with the degree observed in some other TEI studies. To help readers understand size of the effect, we introduce our results in our abstract with this new sentence: We found that chronic and intermittent alcohol-treatment paradigms resulted in opposite changes to intoxication sensitivity of F3 progeny that were only apparent when controlling for yoked trials. 

>>To address your concern that we found opposite effects and only in the F3 generation, we revised the Discussion of our manuscript to elaborate on how other TEI studies have found delayed or distinct effects in each generation for comparison. To address your concerns about methodology, we revised our Methods to clarify all excellent questions raised by reviewer #1. 

We look forward to receiving your revised manuscript.

Kind regards,

Jill C Bettinger, Ph.D.

Academic Editor

PLOS ONE

Reviewers' comments:

Reviewer's Responses to Questions

Comments to the Author

1. Is the manuscript technically sound, and do the data support the conclusions?

Reviewer #1: Partly

Reviewer #2: Partly

2. Has the statistical analysis been performed appropriately and rigorously?

Reviewer #1: Yes

Reviewer #2: Yes

3. Have the authors made all data underlying the findings in their manuscript fully available?

Reviewer #1: Yes

Reviewer #2: Yes

4. Is the manuscript presented in an intelligible fashion and written in standard English?

Reviewer #1: Yes

Reviewer #2: Yes

5. Review Comments to the Author

Reviewer #1: Guzman et al describe two experiments designed to examine the potential for ethanol to generate transgenerational effects on behavioral responses to the effect of ethanol on locomotion in later generations. The development of a protocol to detect true trans-generational effects of the drug would be highly significant, particularly in combination with a genetically tractable organism such as C. elegans. The authors employ two ethanol exposure paradigms, one supported by the literature for generating developmental and withdrawal-associated effects, and the other, an intermittent exposure that is used to minimize potential effects of ethanol on feeding and growth. They examine F1, F2 and F3 generations for altered responses to ethanol, and only detect changes in the F3 generation under each ethanol exposure paradigm. The F3 generation of the chronically-exposed animals have a degree of ethanol resistance relative to paired control animals, whereas the F3 generation of the intermittently-exposed P0 animals have a degree of ethanol hypersensitivity. The effects are subtle enough that statistical differences between worms with an ancestral history of ethanol exposure and those without are only detectable when variation in baseline and day to day variation in behavioral response is considered. The data from each experiment are analyzed in several ways to illustrate the proposed differences in the F3 generation animals.

One issue of significant concern is that effects are only detected in the F3 generation, rather than in the F1, F2 and F3 generations as might be expected with a putative epigenetic effect. That they are not seeing changes in the earlier generations might be considered supporting evidence for these not being epigenetic effects related to the ethanol exposures. The authors should provide reasonable speculation or evidence from the literature to explain why the observed changes in behavioral responses to ethanol are only seen in the F3, not in the F1 and F2 generations, which presumably are also bearing a similar epigenetic change if it was induced in the oocytes (and possibly sperm) of the P0.

>>We also did not expect differences only in the F3 and not in F1 and F2 progeny. As requested, we add discussion on other examples from published TEI studies where effects were found in F3 but not F1 or F2. We also note cases where different effects were found for each F1, F2 and F3 generation. Lastly, we add our own speculation and recommendations for follow up studies to address this unexpected result.

I have several methodological queries and points requiring more detail that are very important for understanding where some of the behavioral variation may be occurring.

>>These are great issues to address. We have edited text of our manuscript to elaborate on and clarify these and additional methodological issues.

1. There appears to be an assumption that the volume of media in the petri plates is identical for every plate, such that 280 µl of ethanol is added to every plate to achieve a final ethanol concentration of 400 mM. If there is any variation in the volume of media, then different plates will have different ethanol concentrations. How are the authors ensuring that the volume of media is the same for each plate? Even if a very accurate method of adding media to the plates is used, media in petri plates often dehydrate over time and at different rates depending on the laboratory conditions and their method of storage. The methods need to be expanded to describe this.

>>To minimize differences in agar volume of assay plates, we only used agar plates that had been poured within 2 days and weighed 18000 mg +/-500 mg (including plastic lid, base and agar). We added this to our revised Methods. We agree that there may still be subtle differences in final concentration of ethanol from plate to plate. This is why we completed all of our behavioral assays with a yoked experimental design. In all cases, we tested control- and EtOH-line worms simultaneously on the same untreated and EtOH-treated plates. If there were subtle plate-to-plate differences, then the two lines would have experienced these together. We have elaborated on this potential problem and our solution in our revised manuscript. 

>>In addition, we revised our Discussion to suggest that future studies may consider adjusting the volume of added EtOH depending on the volume of the plate as inferred through its mass to achieve even more controlled conditions.

2. Related to point 1, is the volume of OP50 culture added to each plate the same? If not, then different plates will have had different volumes of liquid added prior to the 37°C dehydration step.

>>For parental generation, we cultured worms on plates seeded with the same volume (0.5 mL) of OP50 bacterial solution for both untreated and EtOH treated plates. For F1-F3 generations, they were cultured with 0.5 mL OP50. For their behavioral assays, however, all baseline and EtOH treatment assays were conducted on unseeded plates. So OP50 volume was not a factor for F1, F2 and F3 worms during behavioral analyses. We edited our revised Methods to clarify. 

3. The length of time used for the timed egg-laying to produce age-matched progeny should be specified. A shorter window of egg laying will lead to animals that are more closely age-matched.

>>Please note that we performed selections to synchronize developmental age at three phases. First, we synchronized egg laying. After recovery from control or EtOH treated plates, we waited 1 hour before collecting eggs over a 2-10 hour time window. The 1 hour period was chosen as a time point when EtOH was expected to be expelled and/or metabolized by the worm to undetectable levels. Second, we later selected midstage L4 larvae (an approximate 3 hr time window of development) using morphology of the vulva. Third, we further synchronized by selecting day 1-2 adults that harbored between 8-30 eggs. To clarify this important issue, we added a new subsection on animal synchronization in our revised Methods.

4. Related to point 3, the animals used for behavioral testing are described as age-matched day 1-2 adults. There is a significant difference in size of day 1 and day 2 adult worms. Are the worms being compared specifically day 1 adults or day 2 adults or is there a mixture of ages? Importantly, size differences between animals could lead to issues with the method used to track locomotion if body size was not controlled as locomotion is being measured based on area coverage over time.

>>We used a mixture of day 1 and 2 adults because we have used this window in our own on studies on behavioral responses to ethanol in C. elegans without noticing a difference (e.g. Davis et al., 2014; Davis et al., 2015). However, in our current and past studies the developmental age of adults was actually more constrained than 48 hours. We selected adult worms that had at least a row of 8 eggs, which would eliminate early first day adults, but not more than an estimate of 30 eggs. We decided on these exclusion criteria to prevent including worms that have single larvae hatch inside and injure their nervous system which causes them to stop moving during the assay. Such a problem would cause us to erroneously think the worm was more sensitive to ethanol when using locomotion as a behavioral endpoint. Similar exclusion criteria have been used in past studies to focus on healthy motile worms (e.g. Pierce-Shimomura et al., 2001). We did not notice many animals that fell outside of these exclusion criteria in all conditions. We added this information to our new section on animal synchronization in our revised Methods.

5. For the baseline locomotion measurement, the time for how long the worms have been on the plate before the movie of their movement was recorded should be specified. If the recording is immediate, is there concern that the measurement is including the reaction of the worms to just having been moved?

>>Thank you for noticing this important detail. For baseline measurements, we started recording within 90-120 seconds after transfer. This period is after worms suppress their spontaneous reversals for 90 second and display increased displacement during locomotion due to effects of transfer. We have updated our Methods to include this information. 

6. If the worms are moved from seeded plates to unseeded plates for the behavioral assays, are the worms moved from those seeded plates without transferring any bacteria? If so, how? Bacteria that are transferred with the worms will affect the locomotion pattern of worms on a plate that lacks bacteria elsewhere.

>>Thank you also for noticing this detail. We now explain in Methods that we picked worms to a separate unseeded plate to allow bacteria on their bodies to fall off before transferring them to the assay plate. Plates with any visible food were excluded from our study before analysis and with experimenter blind to treatment group. 

7. Why are there different numbers of 1st (n=32), 2nd (n=27), and 3rd (n=37) generation yoked trials? This is also the case for the intermittent exposure. The methods suggest that behavioral testing was performed at each generation of a lineage, so there should be an equal number of trials across the generations. Also, the text references N=38 for 3rd generation analyses but only 37 trials are listed in the supplementary data for F3 trials.

>>We originally aimed to collect equal amounts of data for each generation. However, due to time limitations and logistical issues over each 2-week cohort, we could not achieve this goal. Please note that in parallel to the animals assayed, we propagated many more worms for each generation and selected a random subset to assay for behaviors. Also, thank you for finding the typo, which we fixed to 37. 

Minor issues:

line 81 “a” mystery

line 88 EtOH should be written out in full when first used

line 125 “may be benefit”, delete “be”

line 271 “trial” should be plural

line 430 perhaps add the word “it” to “we could not observe (it) in our experiments”

Throughout: “fertilization” should be used rather than “conception”

>>Thank you for finding these issues. We have corrected these and a few others in our updated manuscript.

Reviewer #2: This is an interesting study describing the transgenerational impact of prior ethanol exposure on locomotion in subsequent generations. While the findings are interesting, the characterization of these findings is very limited. In particular, the metric of area covered is not detailed enough to truly represent intoxication. Other metrics could be measured such as average speed or amplitude or many other measurements of locomotory behavior. 

>>Thank you for your interest in our study. We hope to expand on this in the future.

>>For our study, we initially tracked speed using image analysis of worm centroids. However, we found that centroid tracking presented several problems for such a large study (>440 assays) that were overcome by the area-covered measure instead. Problems with using centroid tracking of speed included: 1) worm speed may artifactually increase when the worm centroid jitters for a worm that is hardly moving, 2) if the worm bent into a loop, the algorithm inappropriately locates the centroid in the open loop rather than on the worm’s body, 3) centroid tracking required expensive software (ImagePro Plus) whereas area covered measurements could be calculated with free FIJI software by multiple users in parallel on different computers, 4) centroids of individual worms would often appear to collide when worms came into contact. To distinguish the multiple worms, we were required to have students laboriously select centroids by hand over many frames, 5) user-defined centroids introduced more subjective decisions into a process that ideally should be as objective as possible, 6) due to time taken to retrack centroid by hand, we found that image analysis using “area covered” was typically 10x faster (2 minutes per assay) than using centroid tracking (20 minutes per assay) and required far less manpower.

>>We also found that the area covered measure offered several advantages over centroid tracking. 1) if worms collided briefly or stayed next to each other for extended periods waving the tip of their heads, the area covered algorithm captured both head and full body movement of the adjacent worms because worms in our assays did not crawl on top of each other, 2) the area covered measure captured differences in body posture – the perceived area of a flaccid worm is less than for a sober worm with a wider S-shaped posture. The original paper on behavioral responses of C. elegans to EtOH demonstrated this usefulness of using animal postures as qualitative behavioral endpoint over centroid tracking in Figure 3 (Davies et al., 2003). Our analysis using area covered makes this into a quantitative measure.

>>Lastly, we also found that normalized locomotion values using area covered correlated with centroid speed for a subset of data with high statistical significance (adjusted R2=0.84, p<0.001, n=152). Please see figure below. Thus, we feel justified in using this more convenient measure.

The authors often reference published work but have not done the work in this particular study to truly show that the worms are indeed being intoxicated. Even the term intoxicated is vague and should be replaced by a more apt term. 

>>We agree that worms, and even rodents, likely cannot experience the same effects of alcohol as humans do during intoxication. To be responsive to your advice, we have changed the title of our paper to “Transgenerational effects of alcohol on behavioral sensitivity to alcohol in Caenorhabditis elegans”. However, rather than continuously write out “behavioral responses to acute alcohol treatment” we revised our Abstract to explain that for simplicity we refer to this in shorthand as “intoxication”. This is also to remain consistent with two decades of precedence for C. elegans studies that refer to acute behavioral responses to ethanol as intoxication (e.g. Davies et al., 2003; Crowder, 2004; Mitchell et al., 2007; Feinberg-Zadek et al., 2005; Feinberg-Zadek et al., 2007; Ient et al., 2012; Dillon et al., 2013; Topper et al., 2013; Davis et al., 2014; Davis et al., 2015; Dopico et al., 2016; Scott et al., 2018a; Scott et al., 2018b; and Oh et al., 2019). 

>>In our current study, we found that worms in every generation (F1, F2, F3) and in both lines (control and EtOH) covered less area when treated with EtOH than in baseline condition. This decrease in area covered is explained by the depressive effects of EtOH on locomotion and posture. We believe that this represents a useful behavioral endpoint to model of intoxication using C. elegans. 

Finally, Figure 2 is low quality and should be redone.

>>We apologize for the problematic figure. We have adjusted the graphical design of Figure 2 and its legend in our resubmission. 

6. PLOS authors have the option to publish the peer review history of their article (what does this mean?). If published, this will include your full peer review and any attached files.

Do you want your identity to be public for this peer review? For information about this choice, including consent withdrawal, please see our Privacy Policy.

Reviewer #1: No

Reviewer #2: No

---

## [Decision Letter · Decision Letter 1]

29 Sep 2022

Transgenerational effects of alcohol on behavioral sensitivity to alcohol in Caenorhabditis elegans

PONE-D-22-19245R1

Dear Dr. Pierce,

We are pleased to inform you that your manuscript has been judged scientifically suitable for publication and will be formally accepted for publication once it meets all outstanding technical requirements.

Kind regards,

Jill C Bettinger, Ph.D.

Academic Editor

PLOS ONE

Reviewer #1: All comments have been addressed

Reviewer #2: All comments have been addressed

---

## [Editor Report · Acceptance letter]

6 Oct 2022

PONE-D-22-19245R1 

Transgenerational effects of alcohol on behavioral sensitivity to alcohol in *Caenorhabditis elegans*

Dear Dr. Pierce:

I'm pleased to inform you that your manuscript has been deemed suitable for publication in PLOS ONE. Congratulations! Your manuscript is now with our production department. 

Kind regards, 

on behalf of

Dr. Jill C Bettinger 

Academic Editor

PLOS ONE